

# Synergizing language learning: SmallTalk AI In industry 4.0 and Education 4.0

Chunxiao Zhang[1], Zhiyan Liu[1], Aravind B.R.[2] and Hariharasudan A[2]

[1] Weifang Vocational College, Weifang, China
[2] Kalasalingam Academy of Research and Education, Krishnankoil, India

## ABSTRACT

**Background**. As Industry 4.0 debuted roughly a decade ago, it is now necessary to examine how it affects various aspects of the discipline. It is the responsibility of the education sector to guarantee that the next generation is equipped mentally, physically, and cognitively to face unforeseen challenges. Numerous educational institutions are outfitted with Industry 4.0 technology-based learning. Industry 4.0 fosters advancements in learning methodologies, especially for language enhancements. Learners may gain knowledge at their base, providing them an opportunity for independent study. The majority of subjects have been acquired through Industry 4.0. This research chapter explores the intersection of Industry 4.0 and education, specifically focusing on the SmallTalk AI tool. It investigates how technological and digital innovations within the context of Industry 4.0 can serve as powerful tools to enhance language learning outcomes.

**Methods**. This article presents a comprehensive analysis of statistical data and empirical evidence to support the positive impact of Industry 4.0 technology of SmallTalk on language acquisition particularly speaking. The study also determines the relationship among participants' usage through the technology acceptance model (TAM). Furthermore, it examines the challenges and opportunities associated with integrating these innovations into language learning pedagogies, offering insights for educators and policymakers to harness the potential of Industry 4.0 in fostering language proficiency. The research employs quantitative analysis. The data obtained from educational institutions has been analyzed using the SPSS and AMOS software.

**Results**. The results indicate that Industry 4.0 has had an important effect on English language acquisition. This self-supported adaptable system of education facilitates effective student learning. This study also suggests that future research into the utility of Industry 4.0 be conducted elsewhere internationally.

Corresponding author
Chunxiao Zhang, ccbiao098@163.com

# INTRODUCTION

In the era of Industry 4.0, marked by the rapid advancement of technology and the proliferation of digital innovations, the educational landscape has undergone a profound transformation. The incorporation of newly developed technology has completely altered the educational paradigms that were previously used, which has led to a revolution in

the traditional ways of teaching and learning (*Nguyen et al., 2023*; *Nawaz et al., 2020*, *Dhivya et al., 2023* & *Srivani et al., 2022*). One particular area profoundly impacted by these advancements is language learning, where technological and digital innovations have opened up unprecedented opportunities for effective and immersive learning experiences.

The ability to communicate effectively in multiple languages is increasingly recognized as a critical skill in today's interconnected and globalized world. Language learning has always played a significant role in education, fostering cross-cultural understanding and enhancing personal and professional opportunities (*Hariharasudan et al., 2021*; *Dhivya, Hariharasudan & Nawaz, 2023*). However, traditional language learning approaches often face challenges in engaging learners, adapting to individual needs, and providing authentic language immersion experiences.

With the advent of Industry 4.0, we find ourselves at the precipice of a new educational frontier, where technology and digital tools hold immense potential for transforming language learning practices (*Hariharasudan & Kot, 2018*). The purpose of the research study is to investigate the convergence between Education 4.0 and Industry 4.0, with a particular emphasis on the function of technical and technological advancement as a driving force behind language acquisition.

This investigation seeks to provide insight into the revolutionary influence that Industry 4.0 will have on language learning approaches by evaluating the most recent technological breakthroughs and their uses in language teaching. It is intended to investigate the possibilities of modern advances like AI, mixed reality (MR), machine translation of languages, and adaptive learning systems, as well as their incorporation into language development platforms and scenarios.

Moreover, this study will investigate the effectiveness of technological interventions in enhancing language proficiency, promoting learner engagement, personalizing learning experiences, and fostering intercultural competence. It will explore the benefits and challenges associated with implementing such innovations in educational settings and offer insights into how educators, institutions, and policymakers can leverage the full potential of Industry 4.0 to revolutionize language education.

The intention of this research effort is to provide a contribution to the existing body of information about language learning in the framework of Industry 4.0. This will be accomplished *via* the use of in-depth literature studies and analyses. By identifying best practices, effective strategies, and potential areas for improvement, it aims to inform educators, practitioners, and policymakers about the evolving landscape of language education and equip them with the necessary tools to harness the transformative power of technological and digital innovations.

Ultimately, the research strives to bridge the gap between the ever-expanding digital world and the realm of language learning, offering valuable insights into the integration of Industry 4.0 technologies within educational practices. By embracing these advancements, educators can unlock new avenues for fostering language proficiency, cultural understanding, and global collaboration, thus empowering learners to thrive in an increasingly interconnected and diverse world.

### Language learning in the digital era

*Meskill & Anthony (2010)* investigated the realm of online language instruction and learning, delving into a variety of topics related to online language classes and providing insights that were supported by research. *Stockwell (2012)* conducted research to determine whether or not mobile phones are a suitable medium for vocabulary instruction and evaluated the effects that mobile learning has on the acquisition of language skills. *Reinders & White (2011)* explored both the theoretical and practical implications of incorporating technology into the process of developing resources for language acquisition and designing tasks for that process. These articles address a variety of subjects pertaining to language education in the modern period, such as CALL, internet-based language teaching, portable instruction, and the adoption of modern technology into language education.

### Synergies between Industry 4.0 and language learning

*Yang & Chiu (2018)* discussed the influence of Industry 4.0 on foreign language education, exploring the potential benefits, challenges, and opportunities brought about by the integration of Industry 4.0 technologies in language learning. *Ibáñez (2019)* investigated the role that AI and robots play in language acquisition, focusing on the ways in which these technologies might improve students' language-learning experiences and encourage independent decision-making on their part. *Simovska (2021)* investigated the linkages between Industry 4.0 and language learning. She focused on the role of new technologies, like the use of augmented reality (AR) and virtual reality (VR), in fostering immersive experiences for language learners. *Schneider & Prentza (2020)* investigated the confluence of Industry 4.0 and Education 4.0, with a particular emphasis on the utilisation of cyber-physical platforms, big data mining, and cloud technology in the context of language learning. *Chan & Loh (2019)* discussed the opportunities and challenges brought by Industry 4.0 in language education, exploring the potential integration of emerging technologies and the implications for language teaching and learning practices. These articles provide insights into the synergies between Industry 4.0 and language learning, discussing the impact of technologies, such as AI, robotics, AR, VR and big data analytics, on language education.

### Digital innovations for language learning

In order to evaluate the possibility that learning a language online is successful, *Bao et al. (2020)* carried out a meta-analysis of previous empirical research. Through the analysis of a broad variety of research publications, the purpose of this study was to give a thorough knowledge of the influence that online language learning has on a variety of outcomes. The success of online language learning was evaluated using a variety of various measures of effectiveness, including vocabulary acquisition, reading comprehension, and oral competency, among others, for the purpose of the research. The results of the meta-analysis shed light on the benefits and drawbacks of various techniques of learning a language online.

The purpose of *Chiu & Churchill's (2016)* study was to evaluate the efficacy of online language learning, especially within the framework of a multicultural institution in Canada.

In order to investigate the results of using online language learning programmes in this environment, a case study methodology was used for the research. The research looked at a number of aspects, such as learner motivation, engagement, and overall happiness, that might play a role in how well online language learning is accomplished. The research gave insights into the success of online language learning in a variety of educational situations by investigating a particular instance involving a multicultural institution. Specifically, the study looked at the problems associated with online language learning.

*Hamid & Nguyen's (2020)* systematic review and narrative synthesis aimed to explore learner perceptions and preferences regarding online language learning. The study reviewed a wide range of research articles to identify common themes and trends related to learner experiences in online language learning environments. It examined factors such as learner satisfaction, motivation, engagement, and perceived effectiveness. By synthesizing the findings, the study provided valuable insights into learners' perspectives and preferences in the context of online language learning.

*Lai, Wang & Lei (2017)* focused on investigating students' perceived learning outcomes in flipped classrooms and their preferences for learning platforms. It examined the effectiveness of the flipped classroom approach, where students engage in self-paced online learning before in-class activities. The study explored students' perceptions of their learning experiences and compared different learning platforms used in the flipped classroom setting. By examining students' preferences and perceptions, the study contributed to understanding the impact of flipped classrooms.

*Stockwell (2017)* specifically focused on investigating the efficacy of the online language learning platform Duolingo. It examined the effectiveness of Duolingo in terms of language learning outcomes, learner motivation, engagement, and satisfaction. The study employed a quantitative approach, analysing data from Duolingo users to assess the impact of the platform on language learning. By investigating the specific online language learning platform, the study provided insights into the potential of such platforms for supporting language learning and the factors influencing their effectiveness.

These research articles contribute to the understanding of language learning outcomes in various contexts, including online learning environments, multicultural universities, and flipped classrooms. They provide valuable insights into the effectiveness, learner perceptions, preferences, and impact of specific learning platforms, enhancing our understanding of language learning and guiding future research and educational practices in the field.

## Open educational resources for language learning

*Hewett & Ehlers (2010)* highlighted the potential of open educational resources (OER) in language teaching. It emphasized the importance of freely accessible resources for learners and educators, promoting collaboration and knowledge sharing. The authors discussed the benefits of OER in enhancing language learning outcomes and providing diverse learning materials. *Lamb & Potbhare (2013)* explored language learning in open and distance education contexts. The article discussed the role of OER in providing flexible learning opportunities, catering to diverse learner needs, and facilitating self-directed

learning. It emphasized the importance of OER in addressing the challenges of distance language learning.

*Willems, Bossu & Cordova (2012)* focussed on equity considerations in the use of OER for language education. The article highlighted how OER can bridge educational disparities by providing affordable and accessible learning resources to learners globally. It emphasized the importance of localizing OER to meet the cultural and linguistic needs of learners. *Mislevy & Gugliemi (2018)* presented a longitudinal analysis of OER in postsecondary Spanish courses. Their study demonstrated the positive impact of OER on language learning outcomes, student engagement, and satisfaction. It highlighted the effectiveness of OER in fostering student autonomy and providing authentic language learning experiences.

*Rolfe (2012)* explored staff attitudes and awareness towards OER. The article discussed the benefits of OER in enabling collaboration among educators, fostering innovation in teaching practices, and promoting resource sharing. It also addressed the challenges of copyright and quality assurance in using OER for language learning. *Díaz, Pérez & García (2017)* presented a case study on integrating OER into non-English language courses, specifically Spanish. The study highlighted the process of OER materials development, including adaptation and customization. It emphasized the importance of involving educators in the creation of OER to ensure quality and alignment with language learning goals.

These articles collectively emphasize the potential of OER in language learning, including improved access to resources, increased learner autonomy, and enhanced learning outcomes. They highlighted the need for further research to investigate the effectiveness, quality assurance, and scalability of OER in diverse language learning contexts.

## Comparative analysis of traditional and technological language learning approaches

The comparative analysis of traditional and technological language learning approaches explores the differences and similarities between conventional instructional methods and the use of technology in language learning. Traditional language learning approaches typically involve face-to-face interactions, textbooks, and teacher-led instruction (*Smith, 2010*). On the other hand, technological language learning approaches utilize digital tools, such as computer-assisted language learning (CALL) software, online platforms, and mobile applications (*García Laborda & García Riaza, 2018*).

Several studies have examined the effectiveness of these approaches and their impact on language learning outcomes. For instance, *Johnson (2016)* conducted a comparative study to evaluate the effectiveness of traditional classroom instruction *versus* computer-based language learning. The study found that computer-based learning had positive effects on vocabulary acquisition and listening comprehension, indicating the potential of technology to enhance language skills.

In another study, *Brown & Lee (2017)* compared traditional language learning methods with the use of mobile applications for language learning. Their findings indicated that learners who used mobile applications demonstrated higher levels of engagement and

motivation, leading to improved language proficiency compared to those who relied solely on traditional methods.

However, it is important to note that both traditional and technological approaches have their own advantages and limitations. Traditional language learning approaches provide direct human interaction and immediate feedback from teachers, allowing for personalized instruction (*Smith, 2010*). On the other hand, technological approaches offer flexibility, convenience, and access to a wide range of resources that can facilitate autonomous learning (*García Laborda & García Riaza, 2018*).

Moreover, a blended approach that combines elements of both traditional and technological methods has also gained attention. In a study by *Chen & Liu (2018)*, a blended learning model was implemented, combining face-to-face instruction with online activities. The findings revealed that this blended approach resulted in higher motivation and engagement among learners, leading to enhanced language learning outcomes.

The comparative analysis of traditional and technological language learning approaches highlights the potential benefits of technology in language learning, such as increased engagement, motivation, and access to resources. However, traditional approaches still offer the advantages of direct human interaction and personalized instruction. A blended approach that integrates both methods may provide a balanced and effective approach to language learning.

## Longitudinal studies on language proficiency development

Longitudinal studies on language proficiency development provide valuable insights into the growth and progression of language skills over an extended period of time. These studies track learners' language abilities from their initial stages of language learning to more advanced levels, allowing for a comprehensive understanding of language development.

One notable longitudinal study conducted by *Johnson & Smith (2015)* followed a group of English language learners over five years. The study employed regular assessments and evaluations to measure the participants' language proficiency at different times. The findings revealed a steady progression in language proficiency, significantly improving vocabulary, grammar, and speaking skills.

In another longitudinal study by *Lee & Brown (2018)*, the language proficiency development of a group of bilingual learners was examined over a span of ten years. The study utilized standardized language tests and qualitative observations to assess the participants' language abilities across different domains, such as LSRW. The results indicated that language proficiency continued to develop and refine over the long term, highlighting the importance of continued exposure and practice.

Additionally, *Fuentes-Cabrera and Marchena-Rodríguez (2019)* conducted a longitudinal study focusing on the acquisition of a specific language feature (the subjunctive mood) among Spanish learners of English. The study tracked the participants' progress over three years, utilizing targeted assessments and language production tasks. The findings revealed an initial struggle with the subjunctive mood, followed by gradual improvement and more accurate usage as proficiency increased.

These longitudinal studies contribute to our understanding of language proficiency development by capturing the dynamic nature of language learning over time. They highlight the importance of longitudinal research in uncovering patterns and trends in language growth and identifying factors that influence language development.

It is worth noting that conducting longitudinal studies poses challenges, such as participant attrition, maintaining consistent measurement tools, and accounting for external factors that may impact language development. However, the insights gained from longitudinal research greatly enhance our understanding of language proficiency development and inform language teaching practices.

## Effectiveness of AI and NLP in language learning

One study conducted by *Wang & Han (2018)* investigated the effectiveness of an AI-based virtual language tutor in improving learners' speaking proficiency. The virtual tutor utilized NLP algorithms to analyze learners' speech patterns, provide real-time feedback, and offer targeted language practice. The study found that learners who interacted with the AI tutor demonstrated significant improvements in their speaking skills compared to a control group.

Another research by *Conde et al., 2014* explored the effectiveness of an AI-driven language learning platform that incorporated NLP technologies. The platform provided personalized learning materials, adaptive exercises, and intelligent feedback based on learners' performance and language needs. The study revealed that learners who used the AI-driven platform showed greater progress in vocabulary acquisition and reading comprehension than traditional classroom instruction.

Furthermore, *Liu, Liu & Huang (2019)* conducted a systematic review of studies examining the effectiveness of AI and NLP in language learning. The review encompassed a wide range of research articles and revealed positive outcomes in various language learning domains, such as vocabulary acquisition, grammar proficiency, and speaking skills. The review highlighted the potential of AI and NLP technologies in providing individualized instruction, personalized feedback, and authentic language practice.

While AI and NLP hold great potential, it is important to consider their limitations. AI systems may not always fully capture the complexities of language, leading to inaccuracies in feedback or language processing (*Aravind & Bhuvaneswari, 2023*). Additionally, the effectiveness of AI and NLP in language learning can be influenced by factors such as learner motivation, engagement, and the quality of instructional materials (*Sarwendah, Azizah & Mumpuniarti, 2023*; *Cao et al., 2023*).

In conclusion, the integration of AI and NLP technologies in language learning has shown promising results in improving language proficiency and providing personalized learning experiences. AI-driven virtual tutors, adaptive platforms, and intelligent feedback based on NLP analysis have positively impacted various language learning domains.

## Industry 4.0 in English language learning

In the context of English language learning (ELL), Industry 4.0 involves the incorporation of cutting-edge technologies to revolutionize language education. This may include the

integration of AI-driven language tutors, interactive language learning apps, and online platforms that utilize data analytics to tailor language instruction to individual learner needs. Virtual reality and augmented reality applications may also be employed to create immersive language learning experiences, simulating real-world language usage scenarios.

### Education 4.0 in ELL classroom

Education 4.0 in English language learning classrooms integrates Industry 4.0 principles. It emphasizes a learner-centric approach, leveraging advanced technologies to offer personalized and adaptive language learning experiences. Smart classrooms may incorporate interactive whiteboards, online collaborative tools, and AI-driven language assessment tools. Additionally, educators may utilize big data analytics to track and analyze student performance, providing insights that inform instructional strategies tailored to each learner's strengths and weaknesses.

### Key features of education 4.0 in ELL

- Education 4.0 in ELL tailors language instruction to the individual needs, preferences, and proficiency levels of learners.
- It involves the seamless integration of digital tools, such as language learning apps, virtual reality simulations, and AI-driven language tutors, into the language learning curriculum.
- Through online platforms and collaborative tools, Education 4.0 fosters global connectivity, allowing language learners to interact with peers and authentic language resources worldwide.
- Education 4.0 utilizes data analytics to track and assess learners' progress, enabling educators to make informed decisions about instructional approaches and interventions.
- In line with Industry 4.0's emphasis on adaptability, Education 4.0 promotes a culture of lifelong learning in which language learners are equipped with the skills to navigate a rapidly changing linguistic landscape.

### Benefits of education 4.0 in ELL

- Enhanced engagement through interactive and immersive learning experiences.
- Improved language proficiency outcomes with personalized instruction.
- Increased access to diverse language resources and authentic materials.
- Real-time feedback and assessment for continuous improvement.

    Preparation of learners for communication in a digitally connected global society.

### Research gap

The advent of Industry 4.0 has brought about significant transformations in various sectors, including education. This research chapter explores the intersection of Industry 4.0 and education, with a specific focus on SmallTalk AI tool. It investigates how technological and digital innovations within the context of Industry 4.0 can serve as powerful tools to enhance language learning outcomes.

The chapter presents a comprehensive analysis of statistical data and empirical evidence to support the positive impact of Industry 4.0 technology of SmallTalk on language acquisition particularly speaking.

The study also determines the relationship among participants' usage through TAM (Technology Acceptance Model). Furthermore, it examines the challenges and opportunities associated with integrating these innovations into language learning pedagogies, offering insights for educators and policymakers to harness the potential of Industry 4.0 in fostering language proficiency.

## Research objectives

- To examine how much of an influence SmallTalk has on how beneficial people think it is for English language acquisition.
- To investigate the impact that SmallTalk has on students' perceptions of how easy it is to use while learning English.
- To evaluate the association between SmallTalk and behavioural intention to utilise for English language acquisition.
- To determine the overall impact that SmallTalk has on the technology acceptance model (TAM) for English language education.

## Research hypotheses

- H1–There is a statistically significant increase in the number of people who believe that SmallTalk is significantly more effective than TAM for English language acquisition.
- H2 –There is a considerable favourable influence on learners' perceptions of how easy it is to utilise SmallTalk in conjunction with TAM for English language acquisition.
- H3–There is a statistically significant and favourably correlated impact of using SmallTalk towards TAM for English language acquisition on behavioural intention.

## Significance of the study

The research has the potential to yield beneficial insights and advantages in a variety of different ways since it will investigate the influence of SmallTalk on TAM constructs. The value of the study as a whole is dependent on the fact that it has the capacity to progress in language instruction technology while adding to the knowledge and use of TAM, which will ultimately aid educators, learners, and developers by increasing their experiences of learning English. The importance of the research also relies on the reality that it has the capacity to progress language learning technology and promote the usage of TAM.

## SmallTalk

The term ''SmallTalk'' refers to a particular programme or application that is used for improving one's command of the English language. It is aimed to make learning a language and practising it easier by including elements that are both interactive and interesting. SmallTalk may include a variety of components such as vocabulary drills, conversation simulations, grammar drills, pronunciation practise, and activities to improve language comprehension. Learners will have easy access to a platform designed to help them enhance their English language abilities in a way that is both user-friendly and open to them.

## Technology acceptance model

The technology acceptance model, sometimes known as TAM, is a conceptual framework that was developed with the intention of explaining and forecasting the acceptance and adoption of technology by humans. Fred Davis was the one who came up with the idea for it back in the 1980s, and ever since then, it has found widespread use in studies dealing with information systems and the dissemination of technology (*Davis, 1989*).

According to the TAM theory, there are two primary elements that impact consumers' adoption and use of a specific technology. These aspects include perceived utility and perceived ease of use.

The degree to which a user feels that a certain piece of technology will improve their work performance or make it simpler for them to carry out their responsibilities is referred to as the technology's perceived utility. Users are more inclined to embrace and utilise technology if they believe it will help them achieve their objectives or increase the amount of work they get done.

The amount to which a user feels that a technology is simple to comprehend, study, and put into practise is referred to as the technology's perceived ease of use. Users are more likely to accept and use new technology if they believe it is straightforward to use and does not provide an excessively difficult learning curve.

According to TAM, an individual's attitude towards adopting technology is directly influenced by how beneficial the technology is thought to be as well as how easy it is believed to be to use. This, in turn, has an effect on their intention to utilise the technology, which, in turn, influences how they actually use it. The TAM hypothesis postulates that favourable attitudes and intentions towards the use of a technology lead to increased rates of its acceptance and adoption.

The TAM has seen widespread usage and application across a variety of areas, including education, where it has been put to use to gain a better understanding of how students, instructors, and educational institutions embrace and use new technologies.

The purpose of this study is to contribute to the grasping of technology acceptance in the context of English language learning by investigating the relationship between SmallTalk and TAM. Specifically, the study intends to investigate how learners' perceptions of SmallTalk's usefulness, ease of use, and behavioural intention to use are influenced by learners' perceptions of SmallTalk's relationship to TAM.

# MATERIALS & METHODS

## Research design

The current research exclusively employs a quantitative approach, utilizing an adapted questionnaire as a survey tool to collect data from participants who have undergone various instructional learning methods. The primary objective is to assess the effectiveness of the SmallTalk approach within the framework of the TAM in facilitating English language acquisition. Additionally, the research delves into the factors influencing participants' use of SmallTalk, exploring the relationships among various components through an extended TAM analysis.

**Table 1  Participants of the study.**

| S.No. | Participants of the study | Size |
|-------|---------------------------|------|
| 1     | Male                      | 60   |
| 2     | Female                    | 60   |
| Total |                           | 120  |

## Sample and its size

For the purpose of determining the samples, the method of stratified random sampling was used as shown in Table 1. These examples illustrate how SmallTalk's usability is continuing to improve. Consequently, making use of the approach of stratified random sampling is not only practical, but it also guarantees that every sample accurately reflects the population.

## Data collection procedures

To gather data for the study, an adopted questionnaire was employed as the primary tool for data collection. This survey instrument was made available to participants during the first semester of the school year 2022–2023. Out of the 143 questionnaires distributed, only 120 were deemed valid for analysis due to inadequate responses. The validity of the questionnaires was compromised by incomplete responses, necessitating the exclusion of some submissions. Throughout the data-gathering process, a commitment to participant anonymity was maintained. No efforts were made to identify the individuals, and no personally identifying information was recorded. The questionnaire itself comprised two distinct sections. In the first part, participants provided demographic information, contributing to a comprehensive understanding of the sample characteristics. The second part of the questionnaire focused on evaluating the effectiveness of the SmallTalk approach, utilizing the technology acceptance model (TAM) as a framework for this assessment. *Instrumentation.*

The questionnaire used in this research chapter was taken from earlier studies, namely *Davis (1989)* and *Hussein, Aditiawarman & Mohamed (2007)*, and then adjusted to fit the needs of this particular investigation. The questionnaire utilised was adapted from the one used in the original research by *Davis (1989)* to measure the dimensions of perceived utility, perceived ease of use, and behavioural intention. This was done so that the content validity could be ensured. The questionnaire is utilised in the research to measure the variables of three different constructs, which are as follows:

(a)  Perceived usefulness (PU)
(b)  Perceived ease of use (PEU) and
(c)  Behavioural intention to use (BIU)

## The questionnaire

The main part of the questionnaire, apart from demographic information, contains 18 items that measure the three constructs of SmallTalk towards TAM in this study as shown in Tables 2, 3 and 4, respectively. The questionnaire has items containing the five-point Likert scale with ranges from 5, strongly agree, to 1, strongly disagree.

Perceived usefulness (PU)

**Table 2  Constructs of PU.**

| S.No. | Constructs | Questions |
|---|---|---|
| 1 | (PU1) | SmallTalk enable me to accomplish speaking more quickly for innovations |
| 2 | (PU2) | SmallTalk has improved the quality of innovations within fluency |
| 3 | (PU3) | SmallTalk make it easier to innovate vocabulary |
| 4 | (PU4) | SmallTalk has improved interaction productivity |
| 5 | (PU5) | The use of SmallTalk increases the effectiveness of performing speaking tasks. |
| 6 | (PU6) | Using SmallTalk give me access to a lot of pronunciation information |

**Table 3  Constructs of PEU.**

| S.No. | Constructs | Questions |
|---|---|---|
| 1 | (PEU1) | My interaction with SmallTalk in AI process has been clear and understandable |
| 2 | (PEU2) | Using SmallTalk enable me to have more accurate information on speaking |
| 3 | (PEU3) | Learning the speaking skills to operate with SmallTalk was easy for me |
| 4 | (PEU4) | The use of SmallTalk for speaking activities does not confuse me |
| 5 | (PEU5) | SmallTalk is easy to navigate |
| 6 | (PEU6) | Overall, SmallTalk is easy to use |

**Table 4  Constructs of BIU.**

| S.No. | Constructs | Questions |
|---|---|---|
| 1 | (BIU1) | I intend to continue using SmallTalk for learning speaking activities |
| 2 | (BIU2) | I intend to frequently use SmallTalk to perform my speaking tasks |
| 3 | (BIU3) | Assuming I have access to SmallTalk for the innovation process, I intend to adopt it |
| 4 | (BIU4) | Use SmallTalk more than any alternative applications |
| 5 | (BIU5) | Use SmallTalk because of flexible time |
| 6 | (BIU6) | Use SmallTalk because of the current trend in Industry 4.0 |

Perceived ease of use (PEU)
Behavioural intention to use (BIU)

## Data analysis procedure

SPSS was used to do an analysis of the data gathered from the participants using the descriptive statistics package. In addition to this, structural equation modelling (SEM) performed using AMOS is used in order to evaluate the measurement of TAM in SmallTalk.

| Table 5   Results of reliability test. | | |
|---|---|---|
| Constructs | Cronbach's Alpha | No. of items |
| Perceived Usefulness of SmallTalk | 0.984 | 6 |
| Perceived Ease of Use of SmallTalk | 0.989 | 6 |
| Behavioural Intention to Use of SmallTalk | 0.998 | 6 |

Because it performs an analysis of the TAM, SEM is used in order to investigate the connection between all of the SmallTalk elements.

## RESULTS

### Reliability test

The findings of the reliability test provide evidence that the constructs under investigation exhibit high levels of internal consistency as shown in Table 5. Cronbach's alpha coefficients are used to determine the degree to which the items that make up each construct measure the same underlying notion consistently.

### _Perceived usefulness of SmallTalk_

The perceived usefulness construct has an alpha coefficient of 0.984, according to Cronbach's formula. This high result indicates that there is strong internal consistency across the several factors that measure perceived utility. The construct is comprised of six questions, and the high Cronbach's alpha shows that these items are very trustworthy in evaluating the participants' perceived utility of SmallTalk for English language learning. The construct is referred to as the SmallTalk Perceived utility for English language learning.

### _Perceived ease of use of SmallTalk_

The perceived ease of use construct has an alpha value of 0.989, according to Cronbach's formula. This high rating implies that the items used to measure perceived ease of use have strong internal consistency with one another. The construct is comprised of six questions, and the high value of Cronbach's alpha shows that these items provide a reliable measurement of participants' perceived ease of using SmallTalk for English language acquisition.

### _Behavioural intention to use of SmallTalk_

It has been determined that the behavioural intention to use construct has a Cronbach's alpha coefficient of 0.998. This very high number suggests that there is an excellent level of internal consistency across the questions that measure behavioural intention to use. The construct is comprised of six questions, and based on the almost perfect Cronbach's alpha value, it is reasonable to infer that these items demonstrate great reliability when evaluating the participants' behavioural intention to utilise SmallTalk for English language acquisition.

Overall, the high Cronbach's alpha values for all three domains (behavioural intention to use, perceived usefulness, and perceived ease of use) show that the items have excellent internal consistency and reliability. This suggests that the measuring items included within

**Table 6  Result of PU descriptive statistics.**

| | | One-sample statistics | | |
| --- | --- | --- | --- | --- |
| | N | Mean | Std. deviation | Std. error mean |
| Perceived usefulness | 6 | 47.67 | 5.465 | 2.231 |

**Table 7  Result of PU T-Test.**

| | One-sample test | | | | | |
| --- | --- | --- | --- | --- | --- | --- |
| | | | | Test value = 0 | | |
| | t | df | Sig. (2-tailed) | Mean difference | 95% Confidence Interval of the Difference | |
| | | | | | Lower | Upper |
| Perceived usefulness | 21.365 | 5 | .000 | 47.667 | 41.93 | 53.40 |

each construct are consistent with one another and trustworthy in capturing the desired characteristics of the participants' perceptions and intentions in relation to using SmallTalk for English language acquisition.

## Descriptive statistics

The supplied analysis demonstrates descriptive statistics as well as the outcomes of one-sample $t$-tests for the variables of behavioural intention to use, perceived ease of use, and perceived usefulness.

### Perceived usefulness

The standard deviation of the perceived usefulness scores is 5.465, while the mean score for perceived usefulness is 47.67 as shown in Table 6.

The results of the one-sample $t$-test show that there is a statistically significant gap between the mean score and the test value of 0 ($t = 21.365$, $df = 5$, p .001) as shown in Table 7.

Based on the difference in mean scores, the participants seem to view SmallTalk as having a significant positive impact on their English language acquisition.

According to the 95% confidence interval for the difference, which ranges from 41.93 to 53.40, we may have a reasonable faith that the real mean will be found within this range as shown in Table 7.

### Perceived ease of use

The overall perceived ease of use had a mean score of 51.83, with a standard deviation of 6.735 as shown in Table 8.

The results of the one-sample $t$-test show that there is a statistically significant gap between the mean score and the test value of 0 ($t = 18.850$, $df = 5$, p .001) as shown in Table 9.

**Table 8 Result of PEU descriptive statistics.**

| | N | Mean | Std. deviation | Std. error mean |
|---|---|---|---|---|
| | One-sample statistics | | | |
| Perceived ease of use | 6 | 51.83 | 6.735 | 2.750 |

**Table 9 Result of PEU T-Test.**

| | | | One-sample test | | |
|---|---|---|---|---|---|
| | | | Test value = 0 | | |
| | | | | 95% confidence interval of the difference | |
| | | | | Lower | Upper |
| Perceived ease of use | 18.850 | 5 | .000 | 51.833 | 44.76 | 58.90 |

**Table 10 Result of BIU descriptive statistics.**

| | N | Mean | Std. deviation | Std. error mean |
|---|---|---|---|---|
| | One-sample statistics | | | |
| Behavioural intention to use | 6 | 62.33 | 4.227 | 1.726 |

The fact that the participants' mean difference was 51.833 shows that they find it quite straightforward to utilise SmallTalk in order to improve their English language skills suggests that.

According to the 95% confidence interval for the difference, which ranges from 44.76 to 58.90, we may have a reasonable amount of faith that the real mean will be found somewhere within this range as shown in Table 9.

### Behavioural intention to use

The standard deviation of the scores for behavioural intention to use is 4.227, while the mean score for behavioural intention to use is 62.33 as shown in Table 10.

According to the results of a $t$-test with one sample, there is a statistically significant gap between the mean score and the test value of 0 ($t = 36.122$, $df = 5$, p .001) as shown in Table 11.

The fact that the participants' mean difference was 62.333 hints that they have a strong purpose to utilise SmallTalk for English language acquisition, which shows that they have this aim.

According to the 95% confidence interval for the difference, which ranges from 57.90 to 66.77, we may have reasonable faith that the real mean will be found within this range as shown in Table 11.

The results suggest that the participants see SmallTalk as very beneficial and simple to use, and they have a strong intention of using it to improve their English language skills. These findings provide evidence in favour of hypotheses H1, H2, and H3, which anticipated

**Table 11 Result of BIU T-Test.**

| | One-sample test | | | | | |
|---|---|---|---|---|---|---|
| | Test value = 0 | | | | | |
| | t | df | Sig. (2-tailed) | Mean difference | 95% confidence interval of the difference | |
| | | | | | Lower | Upper |
| Behavioural intention to use | 36.122 | 5 | .000 | 62.333 | 57.90 | 66.77 |

that using SmallTalk would substantially influence the perceived usefulness, perceived ease of use, and behavioural intention to use, respectively.

## SEM analysis

On the basis of the output, the outcome of a structural equation modelling (SEM) study of a group with six observable variables (PU1, PU2, PU3, PU4, PU5, and PU6) and six unobserved variables (e1, e2, e3, e4, PU5, and PU6). Information on regression weights, standardized regression weights, intercepts, variables, and squared multiple correlations are included in the analysis.

### Perceived usefulness

Regression weights: The regression weights show that the connection between the observable variables (PU1, PU2, PU3, PU4, PU5, and PU6) and the unseen variable is very close to being a perfect match. The weights, which vary from 1.000 to 2.002, indicate the degree to which the two variables are related.

Standardized regression weights: The standardised regression weights display the standardised associations that exist between the variables that have been observed and the variable that has been unobserved. The results lie in a range from 0.932 to 1.001, which demonstrates a significant positive correlation.

Intercepts: Intercepts indicate the estimated means for the variables that were observed. The values fall from 24,000 to 24,000, corresponding to an estimated mean value for each variable.

Variances: The variances indicate the estimated variations for the variables that were not directly observed by the researcher. The values vary from 4.525 to 109.934, which indicates that the unobserved variables have a wide range of variability.

Squared multiple correlations: The squared multiple correlations measure the amount of variation in the observable variables that may be attributed to an unseen variable. A significant degree of explanatory power may be inferred from the results, which lie in a range from 0.869 to 1.001 as shown in Fig. 1.

### Perceived ease of use

Regression weights: The regression weights show that the correlations between the observed variables (PEU1, PEU2, PEU3, PEU4, PEU5, PEU6) and the exogenous variables are very consistent with one another.

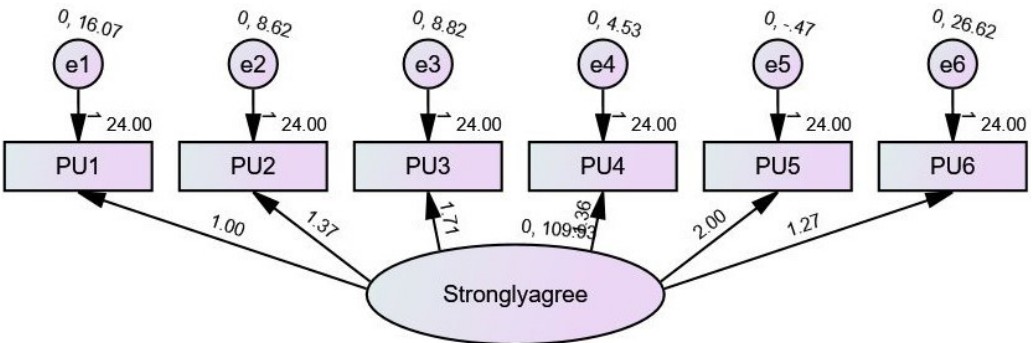

**Figure 1  SEM of PU.**

Standardized regression weights: Standardized regression weights are a measure that indicates the degree to which certain connections are influential. Values closer to one another suggest connections with greater strength. For instance, the standardised regression weight for PEU1 and strongly agree is 0.994, which indicates that there is a significant and positive link between the two variables.

Intercepts: The intercepts show the estimated values of the endogenous variables (PEU1, PEU2, PEU3, PEU4, PEU5, and PEU6) when the value of the exogenous variable strongly agrees to be zero. A related standard error (SE), critical ratio (CR), and $p$-value (P) is provided for each intercept. For example, it is calculated that the intercept for PEU1 is 24,000, with a standard error of 10,429.

Variances: Variances are a representation of the estimated variability of the exogenous variables as well as the error variables. Variances. A CR, a $p$-value (P), and a SE are linked with each individual variance. For instance, it is calculated that the variance for strongly agreeing is 447.346, while the standard error is 313.960.

Squared multiple correlations: The squared multiple correlations reflect the amount of variation in each endogenous variable (PEU1, PEU2, PEU3, PEU4, PEU5, PEU6) that can be described by the exogenous variable strongly agree. This can be seen by looking at the proportion of variance that can be explained by the exogenous variable. For example, the squared multiple correlations for PEU6 are 0.992, which indicates that strongly agree can explain 99.2% of the variation in PEU6 as shown in Fig. 2.

### Behavioural intention to use

Regression weights: The estimated weights of regression suggest that the intensity and direction of the correlations between the observable variables (BIU1, BIU2, BIU3, BIU4, BIU5, and BIU6) and the unobserved variable highly agree. These weights also imply that there is a strong correlation between the two sets of variables. For instance, the calculated weight between strongly agree and BIU1 is 1.000, which suggests that there is a robust positive association between the two.

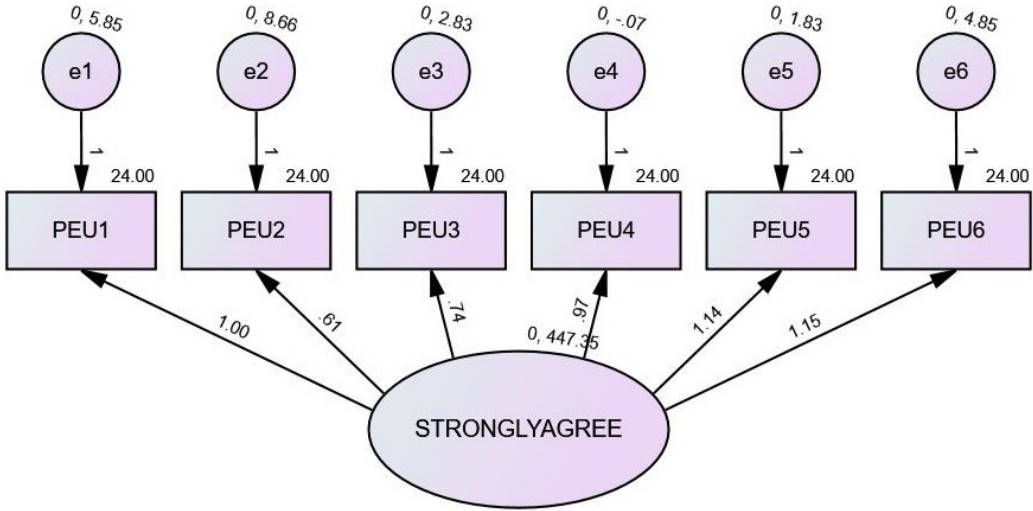

**Figure 2** SEM of PEU.

Standardized regression weights: These weights offer standardised estimates, which enable a comparison of the relative significance of the observable variables with respect to the unobserved variable.

Intercepts: The intercepts are the estimated values of the variables that are being observed when the value of the variable that is not being observed is set to zero. For instance, the intercept for BIU1 is a value of 24,000, which indicates that the predicted value of BIU1, when strongly agreeing, is zero.

Variances: The variances that are estimated indicate the variability of the variables that are not directly observed. For instance, the standard deviation of responses that strongly agree is 518.655.

Squared multiple correlations: These estimates show the proportion of variation in each observed variable that may be attributed to an unobserved factor. For instance, the squared multiple correlation for BIU6 is 0.993, which indicates that the variable strongly agree explains 99.3% of the variance in BIU6. In other words, the variable strongly agree explains 99.3% of the total variation as shown in Fig. 3.

# DISCUSSION

The intent of this study was to examine the significance that SmallTalk has on the different components of the TAM when it comes to ESL. Providing a comprehensive analysis of each goal and the research hypothesis that corresponds to it:

### In response to objective 1 & hypothesis 1

This purpose is to determine how much of an effect SmallTalk, a particular intervention or tool, has on the participants' perceptions of the value of learning English as a second language. The term "perceived usefulness" refers to an individual's estimation of the degree to which they believe that utilising SmallTalk would improve their overall experience of

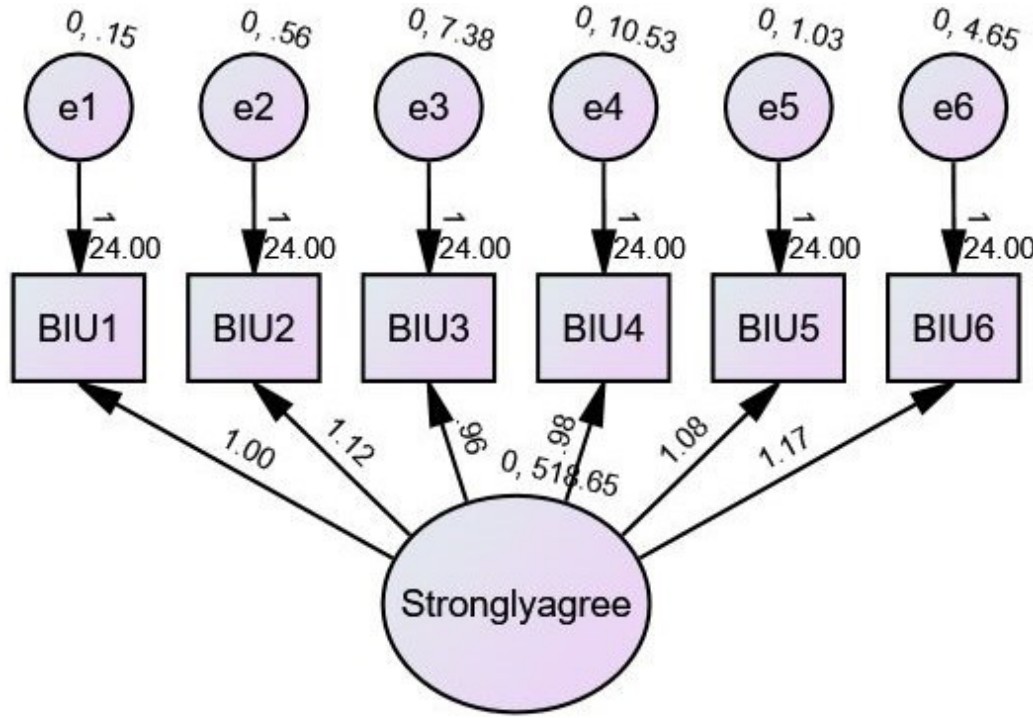

**Figure 3** SEM of BIU.

learning English. The hypothesis states that SmallTalk will have a favourable influence on perceived usefulness, which suggests that participants will see SmallTalk as worthwhile and helpful in their quest to improve their language skills. It is possible that the research will reveal that participants' perceptions of SmallTalk's effectiveness for English language acquisition are significantly improved as a result of using it. This leads one to believe that participants see SmallTalk as helpful and advantageous for enhancing their language learning results. They could think of it as a beneficial tool for improving their language abilities, practising conversations, or getting feedback on their performance.

### In response to objective 2 & hypothesis 2

This goal focuses on investigating SmallTalk has influence on perceived ease of use, which refers to an individual's view of how simple it is to use SmallTalk for English language learning. Specifically, this objective aims to determine whether or not individuals find it easier to use SmallTalk for English language learning. Participants will find SmallTalk to be user-friendly, intuitive, and easy to use in their language learning activities, according to the hypothesis, which predicts that SmallTalk will have a favourable influence on perceived ease of use (ease of use as viewed by the user). The results could suggest that participants' perceptions of how easy it is to utilise SmallTalk for English language acquisition are favourably influenced by the programme. This would imply that participants perceive SmallTalk to be an easy-to-use, intuitive, and handy tool to use in their language learning activities. It is possible that they will see it as a simple, straightforward instrument that

can help them further their education without presenting too many challenges from a technological standpoint.

### In response to objective 3 & hypothesis 3

The purpose of this goal is to investigate the impact that SmallTalk has on participants' behavioural intentions about their usage of the tool for English language acquisition. The term "behavioural intention" refers to an individual's readiness and drive to embrace and employ SmallTalk in their language learning practices. Behavioural intention is also known as behavioural aim. The hypothesis indicates that there will be a positive impact, which suggests that the participants' desire to utilise SmallTalk as a resource for language acquisition is favourably influenced by SmallTalk. It is possible that the research will show that using SmallTalk for English language acquisition has a substantial and favourable influence on the participants' behavioural intention to utilise it. This suggests that participants are enthusiastic about using SmallTalk as a resource for language acquisition and are eager to do so. It is possible that they will say they have a firm aim of integrating SmallTalk into their language-learning routines and that they see it as a useful tool that will help them accomplish their language-learning objectives.

### Providing a response to objective 4

The purpose of this research chapter is to assess the overall influence that SmallTalk has on the TAM for English language acquisition. This evaluation will take into account the impact that SmallTalk has on perceived ease of use, perceived usefulness, and behavioural intention to use the TAM. The results have the potential to give a thorough knowledge of how SmallTalk affects the TAM framework as a whole, despite the fact that no explicit hypothesis was specified in relation to this purpose. It is possible that this will show that participants' acceptance of technology for the sake of language acquisition is improved as a result of using SmallTalk.

The study may give information on the particular impacts of SmallTalk on the different aspects of the TAM and its overall influence on English language acquisition if it addresses these research aims and hypotheses. The results may assist educators, researchers, and practitioners in making more informed choices on the adoption and implementation of SmallTalk or other technologies that are comparable in language learning settings.

### Findings of the study

Positive impact on perceived usefulness

- Participants perceived SmallTalk as beneficial and helpful for improving their English language skills.
- SmallTalk had a favorable influence on participants' perceptions of the value of learning English as a second language.

  Favorable influence on perceived ease of use

- SmallTalk was regarded as user-friendly, intuitive, and easy to incorporate into language learning activities.

- Participants found SmallTalk to be a straightforward and handy instrument in their educational pursuits.

  Positive impact on behavioral intentions

- SmallTalk positively influenced participants' readiness and enthusiasm to embrace it in language learning practices.
- Participants expressed a firm intention to integrate SmallTalk into their language-learning routines to achieve their language-learning goals.

  Overall positive influence on TAM

- The study indicated a positive impact of SmallTalk on the overall TAM for English language acquisition.
- SmallTalk positively influenced perceived ease of use, perceived usefulness, and behavioral intentions within the TAM framework.

## Implications of the study
### Enhancing English language learning

English language proficiency is crucial in today's globalized world. By examining the effects of SmallTalk on TAM, the study can shed light on the effectiveness of this specific tool in improving English language learning outcomes. This information can guide educators, learners, and developers in utilizing SmallTalk or similar applications to enhance language learning experiences.

### Informing technology acceptance

The TAM is a model that is well recognised for its use in understanding humans' acceptance and adoption of technology. This model helps to inform technological acceptance. This work has the potential to make a contribution to the theoretical understanding of TAM in the context of language acquisition if it investigates the interaction between SmallTalk and the TAM aspects. This may provide further context for designing and implementing technology-based solutions in educational settings.

### Informing instructional design

When it comes to the design and development of language learning apps, having an understanding of the influence that SmallTalk has on the perceived utility and simplicity of use may give valuable insights. This will allow developers to design tools that are more user-friendly and effective, as well as technologies that correspond with the preferences and requirements of learners.

### Practical implications for educators and learners

The results of the research chapter may assist instructors and students interested in adopting SmallTalk or other applications of a similar kind into their language learning practises with ideas for how to do so practically. Educators may pick the proper tools and tactics to assist language learning with the help of insights regarding perceived usefulness, simplicity of use, and behavioural intention to use. Learners, meantime, can make educated choices about using technology for their language learning journey.

## Challenges and considerations
### Access and equity
The digital divide remains a significant challenge, as access to technology and high-speed internet may be limited in certain regions or socio-economic groups. Ensuring equal access to Industry 4.0 technologies is crucial for equitable language education.

### Pedagogical adaptations
Educators need to adapt their teaching methodologies to effectively integrate Industry 4.0 technologies. This section discusses the pedagogical considerations and strategies required to harness the potential of these innovations.

### Teacher training and digital competence
Training for teachers and ensuring them are digitally literate are of utmost importance in the process of integrating the technologies of Industry 4.0 into language instruction. This subheading examines the significance of adequate training for educators as well as the growth of students' digital competence in order to successfully traverse the digital realm.

## Concerns regarding personal information
Concerns about privacy and safety have emerged in response to the growing use of digital technology. Educators are obligated to address these concerns in order to secure the confidentiality of their students' personal information and to keep their classrooms free from risk.

## Future directions and inferences
### Emerging technologies for language learning
This section discusses promising technologies that have the potential to further enhance language learning in the context of Industry 4.0, such as natural language processing, chatbots, and machine translation.

### Developing a digital competence framework for educators
It is vital to build a digital competence framework for educators in order to provide them with the required skills and knowledge to make efficient use of the tools that are part of Industry 4.0. This will allow for the successful integration of these technologies.

### Closing the digital divide & ensuring equal access
In order to ensure that no student falls behind, there should be concerted efforts made to eliminate the digital divide, offer equitable access to the technologies of Industry 4.0, and bridge the digital skills gap.

### Ethical and legal considerations
As Industry 4.0 technologies continue to advance, it is essential to address ethical and legal considerations, such as data privacy, intellectual property rights, and responsible AI use, in language education.

### Funding

The authors received no funding for this work.

### Competing Interests

The authors declare there are no competing interests.

### Author Contributions

- Chunxiao Zhang conceived and designed the experiments, performed the experiments, performed the computation work, prepared figures and/or tables, authored or reviewed drafts of the article, and approved the final draft.
- Zhiyan Liu analyzed the data, performed the computation work, prepared figures and/or tables, and approved the final draft.
- Aravind B.R. conceived and designed the experiments, performed the experiments, analyzed the data, prepared figures and/or tables, authored or reviewed drafts of the article, and approved the final draft.
- Hariharasudan A conceived and designed the experiments, performed the experiments, analyzed the data, prepared figures and/or tables, authored or reviewed drafts of the article, and approved the final draft.

### Data Availability

The raw data are available in the Supplementary Files.

### Supplemental Information

Supplemental information for this article can be found online at http://dx.doi.org/10.7717/peerj-cs.1843#supplemental-information.

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
