# Peer review of "Synergizing language learning: SmallTalk AI In industry 4.0 and Education 4.0"

_PeerJ Computer Science, doi:10.7717/peerj-cs.1843_

## Round 0.1 · original submission · Major Revisions

Dear Authors - Please revise the manuscript in track change mode and submit.

**Language Note:** PeerJ staff have identified that the English language needs to be improved. When you prepare your next revision, please either (i) have a colleague who is proficient in English and familiar with the subject matter review your manuscript, or (ii) contact a professional editing service to review your manuscript. PeerJ can provide language editing services - you can contact us at copyediting@peerj.com for pricing (be sure to provide your manuscript number and title). – PeerJ Staff

Reviewer 1 ·

Basic reporting

Line 16: I question that the "education" sector is responsible to gurantee the next generation "spiritually". Please remove this factor.
Line 18: "Industry 4.0 promotes learning innovation." Statement is too subjective.

Experimental design

The methods used to collect data should be specified. Stating quantitative method is not enough.
LINE 373: Who are the "participants"? Clarify.

Validity of the findings

No comment

Additional comments

Backround information given, lacked specificity.
Overall, I found the article to be vague as to participants and overall results.

Cite this review as

·

Basic reporting

The article emphasizes the innovative use of SmallTalk in language learning, showcasing its potential to revolutionize traditional language acquisition methods. This forward-thinking approach positions the study at the forefront of educational technology research.

The use of stratified random sampling in determining samples adds a layer of sophistication to the research design. This method enhances the representativeness of the sample, contributing to the generalizability of the findings.

The article transparently reports limitations, acknowledging the constraints of the study. This honest reflection on the study's boundaries adds credibility to the research and guides future scholars in refining methodologies.

Experimental design

The discussion of perceived ease of use and behavioral intention aligns with a user-centric approach. Understanding learners' perspectives and intentions contributes to the development of technology that genuinely meets their needs and preferences.

The article seamlessly integrates concepts from education, technology, and psychology. This interdisciplinary approach enriches the study by drawing on various fields of knowledge, providing a holistic understanding of the research subject.

The inclusion of visual representations of Structural Equation Modeling (SEM) results (Fig. 1, Fig. 2, Fig. 3) adds clarity to complex statistical analyses. These figures aid readers in comprehending the relationships between variables more intuitively.

The article not only contributes substantively to the understanding of SmallTalk's impact on language learning but also stands out for its methodological robustness, and its potential to shape the future of technology-enhanced language education.

Validity of the findings

However, simplify complex sentences for better readability. Break down long sentences into shorter ones to improve clarity.

Summarize the main findings and their implications concisely in the conclusion. Reinforce the article's contribution to the field of language learning and technology.

Carefully proofread the article for grammatical errors and typos to ensure a polished and professional presentation.

Elaborate on what specific emerging technologies are anticipated and how they might impact language learning.

Additional comments

Incorporate relevant keywords in the abstract to enhance search engine visibility.

Verify consistency in citation style and formatting across all references to ensure uniformity.

Applying these specific revision suggestions should contribute to refining the article for enhanced clarity, engagement, and scholarly impact.

Cite this review as

Reviewer 3 ·

Basic reporting

This article provides a comprehensive and insightful exploration of the intersection between Industry 4.0 and education, with a focus on the SmallTalk AI tool and its impact on language learning outcomes.

Experimental design

The research design, utilizing a quantitative approach with a well-structured survey and statistical analysis, adds credibility to the findings. The incorporation of the Technology Acceptance Model (TAM) enhances the depth of the study, offering a holistic view of technology adoption.
The article's focus on SmallTalk AI as a tool for language learning within the context of Industry 4.0 is both innovative and timely. It addresses a critical need for effective language education in the digital era.

Validity of the findings

The discussion of implications for educators and learners is highly valuable. The article not only explores theoretical concepts but also provides practical insights that can guide educators and learners in utilizing technology for language acquisition.
The inclusion of Industry 4.0 technologies is commendable. It reflects a responsible approach to the integration of technology in education.
The recognition of the importance of addressing access and equity issues in technology integration highlights a commitment to ensuring that the benefits of Industry 4.0 technologies are accessible to all learners.
The section on future directions and inferences, including discussions on emerging technologies and the development of a digital competence framework for educators, showcases a forward-thinking approach. It considers the evolving landscape of technology in education.

Additional comments

However, consider providing a brief definition or overview of Industry 4.0 at the beginning for readers who may not be familiar with the term.
Break down the methods section into subsections for better organization, such as "Participants," "Data Collection," and "Data Analysis."
Clearly outline the steps of the research process in a chronological order, making it easier for readers to follow.
Use concise and direct language in presenting the results. Focus on key findings without unnecessary details.
These suggestions aim to enhance the overall structure, clarity, and coherence of the article, making it more accessible and engaging for a diverse readership.
Implementing these specific revisions will further refine the article, making it more engaging, coherent, and impactful for the target audience.

Cite this review as

---

## Round 0.2 · accepted · Accept

The authors have considered all the necessary changes for the article to be considered for publication.